# Self-Assembling Systems Based on Pillar[5]arenes and Surfactants for Encapsulation of Diagnostic Dye DAPI

**DOI:** 10.3390/ijms22116038

**Published:** 2021-06-03

**Authors:** Anastasia Nazarova, Arthur Khannanov, Artur Boldyrev, Luidmila Yakimova, Ivan Stoikov

**Affiliations:** A.M. Butlerov’ Chemistry Institute of Kazan Federal University, 18 Kremlyovskaya Str., 420008 Kazan, Russia; anas7tasia@gmail.com (A.N.); arthann@gmail.com (A.K.); boldyrev25@gmail.com (A.B.)

**Keywords:** Pillar[5]arene, complexation, DAPI, interpolyelectrolyte, self-assembly, surfactant, visualization system, drug delivery system

## Abstract

In this paper, we report the development of the novel self-assembling systems based on oppositely charged Pillar[5]arenes and surfactants for encapsulation of diagnostic dye DAPI. For this purpose, the aggregation behavior of synthesized macrocycles and surfactants in the presence of Pillar[5]arenes functionalized by carboxy and ammonium terminal groups was studied. It has been demonstrated that by varying the molar ratio in Pillar[5]arene-surfactant systems, it is possible to obtain various types of supramolecular systems: host–guest complexes at equimolar ratio of Pillar[5]arene-surfactant and interpolyelectrolyte complexes (IPECs) are self-assembled materials formed in aqueous medium by two oppositely charged polyelectrolytes (macrocycle and surfactant micelles). It has been suggested that interaction of Pillar[5]arenes with surfactants is predominantly driven by cooperative electrostatic interactions. Synthesized stoichiometric and non-stoichiometric IPECs specifically interact with DAPI. UV-vis, luminescent spectroscopy and molecular docking data show the structural feature of dye-loaded IPEC and key role of the electrostatic, π–π-stacking, cation–π interactions in their formation. Such a strategy for the design of supramolecular Pillar[5]arene-surfactant systems will lead to a synergistic interaction of the two components and will allow specific interaction with the third component (drug or fluorescent tag), which will certainly be in demand in pharmaceuticals and biomedical diagnostics.

## 1. Introduction

The controlled self-assembly of polyfunctional structures into various types of associates opens up wide opportunities for design-targeted drug delivery systems and imaging systems. The implementation of supramolecular chemistry into biomedicine makes it possible to increase the bioavailability, prolong the action of the drug, and prevent its premature degradation, which significantly affects the decrease in the number of side effects and the increase in the therapeutic efficacy of the drug [1,2,3].

The use of Pillar[5]arenes as a macrocyclic component is especially attractive due to the possibility of modification by various functional groups on both sides of the macrocyclic platform [4,5]. Pillar[5]arenes are capable of forming complexes with various biologically significant substrates using their hydrophobic macrocyclic cavity or substituents on both rims of the macrocycle [6,7,8,9,10]. Directed functionalization of Pillar[5]arenes allows one to control such important properties of molecules as bioavailability and water solubility [11,12,13,14,15]. Effective binding of the substrate can be carried out either due to the hydrophobic cavity or due to functional groups [16,17,18,19,20,21,22]. The cooperative action of the cavity and substituents is particularly characteristic during the complexation of charged macrocycles and a substrate, for example, surfactant molecules [23,24,25,26,27]. As a result, 1:1 host–guest complexes are formed [28,29,30,31,32,33,34,35]. In addition, the ability of water-soluble macrocycles, by analogy with non-macrocyclic surfactants, to form various types of associates such as liposomes [36,37,38], micelles [39,40,41], and liquid crystals [42,43], etc., is well known. It attracts great attention due to their potential application in various fields and especially in biomedicine as nanocontainers for medicinal and imaging agents [44,45,46,47,48,49,50,51]. Consequently, it becomes possible to obtain both host–guest complexes and polyelectrolyte aggregates of macrocycle-surfactants by varying the stoichiometric composition of a Pillar[5]arene-surfactant mixture on the analogy of interpolyelectrolyte complexes (IPEC) based on oppositely charged polymers [52,53,54,55,56,57].

The use of IPEC as drug delivery systems or imaging systems is attracting more and more attention of researchers due to the environmental friendliness and safety of such systems [58,59]. Most of the IPEC can be obtained in aqueous media without the use of organic solvents and chemical crosslinking agents by mixing oppositely charged solutions of polyelectrolytes in different ratios. A promising direction in the design of IPEC is the use of macrocyclic compounds, which, due to the presence of a macrocyclic cavity, make it possible to obtain materials with a specified structure and properties [60,61,62]. The interaction of the macrocyclic platform with oppositely charged polyelectrolyte molecules leads to a change in the properties of the supramolecular complex due to the cooperative aggregation of its components [39,63]. The polyelectrolyte, the role of which can be played by various surfactants, can change the aggregation and binding properties of the macrocyclic compound. The macrocyclic molecule, in turn, can fix the polyelectrolyte in a predetermined manner in space.

Summarizing the above, it can be argued that such a strategy for the design of interpolyelectrolyte supramolecular Pillar[n]arene-surfactant systems will lead to a synergistic interaction of two components and will allow specific interaction with the third component (drug or fluorescent label), which will certainly be in demand in pharmaceuticals and biomedical diagnostics. Therefore, the aim of this work was to create supramolecular systems based on oppositely charged Pillar[5]arene and surfactants for encapsulating the dye 4’,6-diamidino-2-phenylindole (DAPI), which is used for fluorescent staining of DNA in fluorescence microscopy [64,65]. We studied the aggregation behavior of synthesized macrocycles and surfactant molecules and investigated the encapsulating ability of mixed IPECs as nanocontainers for a fluorescent label.

## 2. Results and Discussion

### 2.1. Self-Assembly and Characterization of Pillar[5]arene-Surfactant Systems

To obtain stable self-assembling systems based on Pillar[5]arenes and surfactants, the self-assembly of individual components of these systems, namely, macrocycles **1** and **2**, and surfactants sodium dodecyl sulfate (SDS) and dodecyl trimethylammonium chloride (DTAC) was studied. Dynamic light scattering method shows the formation of polydispersity systems with submicron-sized associates for each component at the concentration range 1 × 10^−5^ to 1 × 10^−3^ M (Appendix A). The main feature of surfactants is their ability to form micelles of various sizes. Pillar[5]arenes are convenient macrocyclic platforms that can form a large number of host–guest complexes with various substrates (surfactants, drugs, etc.). Therefore, various nanosystems were created based on the Pillar[5]arenes (liposomes, micelles, vesicles, nanostructured lipid carriers, etc.). In this regard, we made the assumption that by varying the molar ratio in Pillar[5]arene-surfactant systems, it is possible to create various types of supramolecular systems (Figure 1):

I: host-guest complexes at equimolar ratio of Pillar[5]arene-surfactant.

II: stoichiometric interpolyelectrolyte complexes (SIPEC) are self-assembled materials formed in aqueous media by two oppositely charged polyelectrolytes (macrocycle and surfactant micelles) at equimolar ratio of the charged groups.

III: nonstoichiometric interpolyelectrolyte complexes (NIPEC), where the charges of one sign prevail.

Tenfold excess of surfactants gives us mixed systems, stoichiometric interpolyelectrolyte complexes (SIPEC), where ten charged groups of Pillar[5]arene electrostatically interact with ten molecules of surfactant. At the same time, the addition of a significant (100- or 1000-fold) excess of surfactant to Pillar[5]arene leads to the formation of surfactant micelles with included Pillar[5]arene molecules, nonstoichiometric interpolyelectrolyte complexes (NIPEC).

### 2.2. Host–Guest Complexes at Equimolar Ratio of Pillar[5]arene-Surfactant

To describe the formation of various types of systems (I, II, III), we used UV-vis spectroscopy and dynamic light scattering methods. The NMR method, widely used to confirm complexation, is not suitable for these systems, because the complexation process is usually accompanied by the association of macrocycles and surfactants, which leads to broadening of signals in the spectrum and difficulties in their characterization.

At the first stage, the complexing properties of oppositely charged macrocycles and surfactants were studied by UV-vis spectroscopy in water (Figure 2). Solutions of guest-molecules (SDS and DTAC) were added to solutions of compounds **1** and **2** (3 × 10^−5^ M) in a 1:1 ratio, correspondingly. A deviation of the optical density of the mixture (A_complex_) from the additive spectrum of components, (ΣA_mixture_), ΔA = A_complex_ − ΣA_mixture_, was found, which indicates complexation. In both cases, the complexation was accompanied by a hyperchromic effect and a bathochromic shift at 300 nm.

To quantify the complexing ability of compounds **1** and **2** with SDS and DTAC, respectively, the association constant *K_a_* and the stoichiometry of the complexes formed were determined. The stoichiometry of the obtained host–guest complexes 1:1 was determined for both complexes by the Job plots (Appendix A). The method of spectrophotometric titration was used to determine the binding constant. For the Pillar[5]arene 1/SDS and Pillar[5]arene 2/DTAC systems, the Pillar[5]arene concentration remained constant, while the surfactant concentration increased from a ratio of 1:0.5 to 1:10 (Appendix A). The data obtained were processed using BindFit [66], and the association constants of the 1:1 complexes were also calculated (Appendix A). The *K_a_* for **1**/SDS was determined as 1053 M^−1^. The *K_a_* for **2**/DTAC was 102 M^−1^.

### 2.3. Interpolyelectrolyte Complexes (IPECs) Based on Pillar[5]arene and Surfactant

SIPEC and NIPEC can be formed depending on the molar ratio of cationic to anionic groups. In a stoichiometric complex, positive and negative charges are mutually neutralized, and most of the formed particles can precipitate due to colloidal instability. By contrast, in non-stoichiometric complexes, the forming aggregates contains an excess of one of the polyelectrolytes, thereby conferring the formed particle with surface charge and, in consequence, colloidal stability [67].

So, it was necessary to determine the most optimal molar ratio of Pillar[5]arene to surfactant for formation of SIPEC and NIPEC in water solutions. In this regard, we varied macrocycle:surfactant molar ratio (1:10, 1:100, and 1:1000). For each ratio, the influence of the macrocycle concentration was also investigated (c_macrocycle_ = 1 × 10^−6^ ÷ 1 × 10^−3^ M).

To obtain SIPEC, it is necessary to have a minimum 10-fold excess of a surfactant because each of Pillar[5]arenes **1** and **2** have ten charged groups. The amount of surfactant required to obtain SIPEC is also influenced by the basicity and acidity of charged groups and their shielding in self-association processes [68,69]. Indeed, in the case of a 10-fold excess of surfactants in **1**/SDS system at a concentration of 1 × 10^−3^ M and 1 × 10^−4^ M, insoluble precipitates are formed. It indicates the **SIPEC-1** formation (Figure 1, I), when positive and negative charges are mutually neutralized, and most of the formed particles precipitate out due to colloidal instability. Therefore, both the aggregate size formed by **1**/SDS in the case of a 10-fold excess of surfactants and their dispersity depended weakly on the concentration (Appendix A).

The increasing in the SDS content to 1:100 molar ratio at a concentration of 1 × 10^−3^ and 1 × 10^−4^ M results in the fact that precipitates are also formed; however, at a concentration of 1 × 10^−5^ M, a stable colloidal system with a monomodal size distribution of particles is formed (d = 242 ± 9 nm, PDI = 0.13). This behavior of systems is possible with the simultaneous presence of SIPEC and some excess of SDS, which is insufficient to dissolve the precipitate at a relatively high concentration (1 × 10^−3^ and 1 × 10^−4^ M). Dilution of the system leads to dissolution of the precipitate and the formation of **NIPEC-1**, with included SDS molecules. This is confirmed by the value of the ζ-potential—the surface of the aggregates is negatively charged (ζ = −21 mV). Further dilution (1 × 10^−6^ M) leads to destabilization of the system, when the balance of forces of intermolecular interaction and electrostatic repulsion is disturbed, accompanied by coagulation of particles: the particle size and polydispersity of the system increase (d = 1270 ± 632 nm, PDI = 0.82).

A further increase in the SDS concentration (1000-fold of SDS) leads to the formation of **NIPEC-1** (Appendix A), and at a concentration of 1 × 10^−6^ M the most stable particles are formed with a hydrodynamic diameter of 71 ± 2 nm and PDI = 0.16. The negative value of the ζ-potential (ζ = −40.2 mV) of this system indicates that micelles are formed by surfactant molecules with built-in Pillar[5]arene molecules (Figure 1).

The aggregation behavior of **2**/DTAC system is similar to **1**/SDS (Appendix A), where the formation of IPEC occurs by the same mechanism. At 1:10 stoichiometric ratio, the formation of **SIPEC-2** also occurs. However, no visually insoluble precipitate was found. This is possible due to the predominance of soluble IPECs over insoluble ones, or soluble IPECs are products of incomplete interactions [57,70], due to different strengths of polyelectrolytes. In this case, the free units of the initial polyelectrolytes, which did not interact, act as hydrophilic fragments, contributing to the retention of IPEC particles in solution. These assumptions are confirmed by the formation of a stable system with a hydrodynamic particle diameter of 286 ± 13 nm, PDI = 0.23, ζ = −34.7 mV at the concentration of Pillar[5]arene **2** of 1 × 10^−4^ M. The negative value of the ζ-potential indicates the uncompensation of charges on the surface of micelles, which consist mainly of molecules of macrocycle **2**. In the case of a 1000-fold excess of DTAC with respect to Pillar[5]arene **2**, **NIPEC-2** are also formed (d = 170 ± 8 nm, PDI = 0.27, ζ = −21 mV).

Thus, by varying the molar ratio of macrocycle: surfactant, three types of systems were obtained: host–guest complexes at equimolar ratio of Pillar[5]arene-surfactant, SIPEC and NIPEC at surfactant excess.

### 2.4. DAPI Association with IPEC

Synthesized SIPEC and NIPEC due to non-covalent interactions are capable of specific interaction with a third component represented by a fluorescent label (imaging system agent) or a drug (drug delivery system). 4′,6-Diamidino-2-phenylindole (DAPI) is a widespread fluorescent dye that has an affinity for DNA. In addition, DAPI can be used as an antiparasitic, antibiotic, antiviral and antitumor agent. In this regard, we chose DAPI as a guest (third component) to study the interaction with interpolyelectrolyte systems formed by Pillar[5]arene and surfactant molecules. We used UV-vis, fluorescence spectroscopy, nanoparticle trajectory analysis, and dynamic light scattering (DLS) methods for detecting the interaction between the IPEC (SIPEC and NIPEC) and DAPI. Pillar[5]arenes and dye were taken in equimolar ratios (c = 3 × 10^−5^ M). To assess the possible advantages of using macrocycle/surfactant IPEC for DAPI encapsulation, we additionally investigated the interaction of DAPI with each of the IPEC components, in those concentrations and ratios of the starting components that were used to create the IPEC.

#### 2.4.1. UV-Vis Spectroscopy

UV spectrum of **1**/DAPI mixture shows that macrocycle **1**, containing ten ammonium fragments, does not interact with DAPI (Appendix A). The addition of a 100-fold excess of SDS to DAPI results in a hypochromic effect and a bathochromic shift in comparison to the additivity of two spectra of the binary system (Figure 3a). The 1000-fold excess of SDS to DAPI results in only bathochromic shift from 340 to 360 nm (Figure 3c). The addition of DAPI to the **SIPEC-1** (Figure 3b) and **NIPEC-1** (Figure 3d) is accompanied by a bathochromic shift and, additionally for **SIPEC-1**, an increase in absorption of the mixed solution which indicates the formation of large mixed aggregates that are able to scatter light.

There are no changes in the UV spectrum of the DTAC and DAPI mixture (Figure 4b). The addition of DAPI to Pillar[5]arene **2** is accompanied by a hypochromic effect and a bathochromic shift (Figure 4a). The addition of DAPI to **SIPEC-2** also shows a bathochromic shift and a hypochromic effect (Figure 4c). At the same time, for **NIPEC-2**, no visible changes were observed with the addition of the dye (Figure 4d). In the ternary systems described, three types of interactions can be observed: macrocycle–surfactant, macrocycle–dye, and surfactant–dye. Since the surfactant in both studied concentrations does not interact with the dye molecule, and the addition of macrocycle **2** to DAPI leads to a hypochromic effect and a bathochromic shift, it can be assumed that competitive complexation occurs during the formation of ternary systems. For **SIPEC-2**, the decisive factor is the interaction between the macrocycle and the dye, which is probably due to the higher value of the association constant for the **2**/DAPI system than the *K_a_*
**2**/DTAC. At the same time, for **NIPEC-2**, a 1000-fold excess of surfactant leads to the destruction of the **2**/DAPI complex and the release of the dye, which does not interact with surfactant molecules due to electrostatic repulsion. Thus, the analysis of the experimental data showed that the formation of ternary systems occurs, first, due to the electrostatic forces arising between the surfactant and dye molecules.

#### 2.4.2. Dynamic Light Scattering Method (DLS)

DLS data showed different behavior of interpolyelectrolyte complexes (IPEC-1 and IPEC-2) formed by macrocycles **1** and **2** with respect to DAPI (Appendix A). The addition of a dye to **NIPEC-1** only slightly decreased the polydispersity and the size of the ternary system **NIPEC-1**/DAPI aggregates (d = 215 ± 28 nm, PDI = 0.55). A similar behavior is observed for **SIPEC-1**/DAPI: aggregates with a hydrodynamic diameter of 216 ± 2 nm and PDI = 0.06 are formed (Appendix A). DLS data are a good correlate with UV data, where a baseline increase indicated the formation of aggregates (Figure 3b). Thus, the addition of DAPI to **SIPEC-1** (d = 242 ± 9 nm, PDI = 0.13) also slightly decreases the size and PDI of the aggregates formed. According to ζ-potential, the addition of DAPI to **SIPEC-1** (−21 mV) increases the stability of **SIPEC-1**/DAPI aggregates (−36 mV).

For **SIPEC-2**, an opposite behavior is observed: the addition of the dye leads to the enlargement of aggregates and the formation of a system with a wide size range (d = 494 ± 133 nm and PDI = 0.73). At the same time, when DAPI is added to **NIPEC-2**, PDI does not change (PDI = 0.27), and the size of the formed aggregates increases to 683 ± 47 nm (Appendix A). It indicates the aggregation under the action of DAPI. In this case, DAPI molecules stick together **NIPEC-2** aggregates in a certain ordered manner, which only leads to aggregates enlargement without changing the PDI of the system.

#### 2.4.3. Luminescent Spectroscopy, Nanoparticle Trajectory Analysis (NTA) and Molecular Docking

To explain this difference in behavior during the assembly of IPEC with DAPI, we used luminescent spectroscopy and NTA.

Under the experimental conditions similar to UV-vis for Pillar[5]arene **1**, it was shown that the addition of a dye to the macrocycle in the luminescence spectrum of the mixture does not lead to any visible changes (Figure 5a). It is in agreement with the data obtained by UV spectroscopy (Appendix A). The addition of 100-fold excess of SDS to DAPI leads to luminescence flare-up and bathochromic shift (Figure 5b), while with a 1000-fold excess of SDS, only luminescence flare-up occurs (Appendix A). The similar behavior is observed also for the **NIPEC-1**/DAPI system (Appendix A). Oppositely, for **SIPEC-1**/DAPI, system luminescence is quenched (Figure 5c). The enhancement of luminescence is due to the electrostatic interaction of acetamidine fragments in the DAPI with a negatively charged aggregate surface (**NIPEC-1** and SDS micelles). Luminescence quenching is apparently due to the shielding of the dye molecule due to incorporation into the **SIPEC-1** structure due to the π–π stacking of the aromatic rings of the Pillar[5]arene **1** and DAPI (Figure 6). The same type of associates is formed in the **SIPEC-2**/DAPI system, where quenching is observed in the luminescence spectra (Figure 6 and Appendix A).

An unexpected increase in luminescence was observed when a 1000-fold excess of DTAC interacted with DAPI (Appendix A), while a 100-fold excess of DTAC did not cause any changes in the luminescence spectra (Appendix A). Obviously, in these systems, DTAC molecules are in different forms (Figure 6). At a high concentration of DTAC (1000-fold excess), DAPI interacts with DTAC micelles, whereas at a lower concentration (100-fold excess), the surfactant molecules are free and the only one electrostatic type of interaction is impossible in this case (Figure 2). In the case of DTAC micelles, a positively charged micelle due to cation–π interactions concentrates dye molecules on the micelle surface, enhancing luminescence (Figure 6 and Appendix A). By a similar mechanism, DAPI interacts with **NIPEC-2** to induce increased luminescence (Figure 6 and Appendix A).

DINC 2.0 web server [71] was used to confirm the fact that luminescence quenching is associated with the inclusion of a dye into the macrocycle cavity and in the structure of associates **SIPEC-1** and **SIPEC-2** due to the π–π stacking of the aromatic rings of the Pillar[5]arene and DAPI (Figure 7). DINC is a parallelized meta-docking method for the incremental docking of ligands. The strategy of DINC involves incrementally docking overlapping fragments with a growing number of atoms, while maintaining the number of flexible bonds constant during this incremental process [72]. The grid size was set to ~ 30 Å × 30 Å × 30 Å xyz points and grid center was designated at dimensions (x, y, and z): 0.8, 0.8 and −6.4.

Binding energy in the DINC 2.0 server output between Pillar[5]arene **2** and DAPI (host–guest complex) was −6.30 kcal/mol. The obtained docking results exhibited relatively high potential for binding of DAPI with macrocycle **2** (Figure 7a). This indicates that complexation between the compound **2** and DAPI is an energy efficient process. The location of the dye outside the macrocyclic platform turned out to be energetically favorable (−4.1 kcal/mol). This confirms our assumptions about the structure of SIPEC, when the dye molecule is located between macrocycle–surfactant associates (Figure 7b).

Unfortunately, in cases where luminescence increases (SDS/DAPI, DTAC/DAPI, **NIPEC-1**/DAPI, **NIPEC-2**/DAPI), it is not possible to determine the association constant (*K_a_*) due to going beyond the linearity of concentration determination. Despite the fact that luminescence quenching was also observed for the **SIPEC-1**/DAPI system, it is problematic to determine the *K_a_*; SDS contributes to this system, the addition of which to DAPI causes an increase in luminescence. Therefore, the next step is the quantitative characteristics of the interaction of the dye with Pillar[5]arene **2** and **SIPEC-2**. In the luminescence spectrum of pure DAPI (Figure 8a), four main emission bands can be distinguished at 422, 450, 495, and 560 nm. When DAPI interacts with Pillar[5]arene **2**, a nonequivalent change in intensity is observed at bands 422, 450, and 495 nm (Figure 8b). This can be seen more clearly when plotting graphs in the I/I_0_ format (Figure 8c), where I_0_ is the intensity of DAPI emission in the absence of a macrocycle **2**. Thus, for the **2**/DAPI system, three independent emission regions can be distinguished, which indicate the existence of three types of particles with emission in different spectral regions.

To prove that associates of different compositions luminesce in different regions of the spectrum, the analysis of supramolecular aggregates was carried out by the NTA method. Nanosight LM-10 allows to cut off luminescent particles using extinguishing filters. Figure 9 shows the dependence of the size and concentration of nanoparticles in a solution without a luminescence quenching filter (Figure 9a), namely, all particles that are present in the solution. There is also luminescence spectrum of DAPI and DAPI/**2** at a molar ratio of 1:1, which is given for visual display of the type of luminescent nanoparticles (Figure 9c). The average size of all particles in the system is 173 ± 5 nm with a total amount of 12.1 × 10^8^ particles/mL (Figure 9a). A significant decrease in concentration and an increase in the size of the detected particles are observed when using a cutoff filter at 430 nm (Figure 9b).

After using cutoff filters at 430 and 500 nm (Figure 9b,d), the average hydrodynamic particle diameter changes to 244 ± 15 and 203 ± 30 nm, respectively. The increase in error is primarily due to a decrease in signal intensity. It is worth noting that when using a cutoff filter at 430 nm, the particles that we observe when using a 500 nm filter are also recorded. The subsequent study of the **2**/DAPI system at other molar ratios showed that an increase in the content of Pillar[5]arene **2** leads to the normalization and stabilization of the type of luminescent particles. When the molar ratio **2**/DAPI is more than three, there is only one type of luminescent particles in the system in the entire range (Appendix A).

Based on the data obtained, it can be assumed that with an increase in the concentration of **2**, associates of a complex composition are first formed, consisting of DAPI associates and associates of macrocycle **2**. A further increase in the concentration of Pillar[5]arene **2** leads to the destruction of self-associates of the dye and their interaction with the **2**, while maintaining the luminescence of the dye. A significant excess of compound **2** leads to complete solubilization of DAPI with quenching of luminescence [73,74].

This assumption is confirmed (Figure 10a, Table 1) when processing the results obtained by the method of construction of a Scatchard plot [75].

As seen in Figure 10 in the **2**/DAPI system, there are three linearity sections (Figure 10a), and in the **SIPEC-2**/DAPI there are four linearity sections (Figure 10b) from which it follows that the systems have three and four association constants, respectively. For the **2**/DAPI system, we attribute the first association constant to the interaction of compound **2** with its own associate DAPI, since it is in excess. The second constant refers to the true constant of association of individual DAPI molecules with compound **2**. The third constant we refer to is the constant of association of the supramolecular aggregate **2**/DAPI with free molecules of compound **2** along the perimeter. That is why there is no signal in the luminescence spectrum at a wavelength of 500 nm, and consequently, it is impossible to calculate the interaction constant (Table 1).

When switching to the **SIPEC-2**/DAPI system, we observe, first, a general increase in the values of the association constants in the system, and second, an additional association constant. Both effects are explained by the presence of surfactants in the system at a concentration above the critical micelle concentration (CMC) point. That is, the first constant with the maximum energy refers to the solubilization of the dye into the micelle. The subsequent association constants refer to the same processes as in the **2**/DAPI system. The overall increase in the constant value by an order of magnitude or more is explained by the facilitation of the interaction of DAPI and **2** due to the effect of concentration in the micelle. The fact that the association of constants in the **SIPEC-2**/DAPI system to the same processes is confirmed by the absence of the fourth association constant at a wavelength of 500 nm.

## 3. Materials and Methods

### 3.1. General

All chemicals were purchased from Acros (Fair Lawn, NJ, USA), and most of them were used as received without additional purification. Organic solvents were purified by standard procedures. ^1^H NMR spectrum was obtained on a Bruker Avance-400 spectrometer (Bruker Corp., Billerica, MA, USA) (^1^H 400 MHz). Chemical shifts were determined against the signals of residual protons of deuterated solvent (D_2_O, DMSO-*d_6_*). The concentrations of the compounds were equal to 3–5% in all the records. Melting points were determined using Boetius Block apparatus (VEB Kombinat Nagema, Radebeul, Germany). UV-vis spectra were recorded on a Shimadzu UV-3600 spectrometer (Kyoto, Japan). All the aqueous solutions were prepared with the Millipore-Q deionized water (> 18.0 MΩ cm at 25 °C). The size, concentration and movement of nanoparticles were determined using a NanoSight LM10 instrument (Malvern Instruments Ltd., UK) equipped with a C11440-50B CMOS camera with an FL-280 scientific image sensor (Hamamatsu Photonics, Japan) as a detector. Luminescence spectra were recorded on a Perkin Elmer LS 55 (Perkin Elmer, Waltham, MA, USA).

**4,8,14,18,23,26,28,31,32,35-Decakis-[(N-(3′,3′,3′-trimethylammoniumpropyl)carbamoylmethoxy]Pillar[5]arene decaiodide (1)** was prepared by a literature method [76].

**4,8,14,18,23,26,28,31,32,35-Deca(carboxymethoxy)Pillar[5]arene tributylammonium salt (2)** was prepared by a literature method [77].

### 3.2. UV-Vis Spectroscopy

Absorption spectra were recorded on a Shimadzu UV-3600 spectrometer (Kyoto, Japan). Quartz cuvettes with an optical path length of 10 mm were used. Deionized water was used for preparation of the solutions. Absorption spectra of mixtures were recorded after 1 h incubation at 20 °C. Solutions of SDS and DTAC were added to those of compounds **1**, **2** (3 × 10^−5^M) in a 1:10 ratio to study the complexation of Pillar[5]arenes with guests in solvent.

#### 3.2.1. Determination of the Stability Constant and Stoichiometry of the Complex by Spectrophotometric Titration

A 3 × 10^–3^ M solution of SDS (DTAC) (30, 60, 90, 120, 150, 180, 210, 240, 270 and 300 μL) in water was added to 0.3 mL of a solution of **1** (**2**) (3 × 10^−4^M) in water and diluted to a final volume of 3 mL with water. The UV spectra of the solutions were then recorded. The stability constant of complex was calculated by Bindfit [66]. Three independent experiments were carried out for each series.

#### 3.2.2. Job Plots

Job Plots for hosts **1** and **2** and guests SDS and DTAC were determined in deionized water with the ratio varied from 0.6:2.4 to 2.4:0.6. Each of the measurements of the series was performed three times.

### 3.3. Dynamic Light Scattering (DLS)

#### 3.3.1. Particles’ Size

The Zetasizer Nano ZS instrument (Worcestershire, UK) equipped with the 4 mW He-Ne laser (633 nm) was used for the determination of particle size. Measurements were performed at a detection angle of 173° and the software automatically determined the measurement position within the quartz cuvette. Processing of the results was performed by the DTS program (Dispersion Technology Software 4.20). Deionized water was used to prepare the solutions. The concentrations of the compounds **1** and **2** were 1 × 10^−6^ M, 1 × 10^−5^ M, 1 × 10^−4^ M, and 1 × 10^−3^ M. The particle sizes were measured after 1 h mixing. To study the aggregation of Pillar[5]arenes with surfactants, water solutions of SDS and DTAC were added to the solution of compounds **1** and **2** (1 × 10^−3^ ÷ 1 × 10^−6^ M) in chloroform at 1:1, 1:10, 1:100, and 1:1000 ratios. The particle sizes were measured after 1 h mixing. Measurements were determined after 24 and 178 h three times to evaluate kinetic stability.

#### 3.3.2. Zeta Potentials

Zeta (ζ) potentials were measured on a Zetasizer Nano ZS from Malvern Instruments (Worcestershire, UK). Samples were prepared as for the DLS measurements and were transferred with the syringe to the disposable folded capillary cell for measurement. The zeta potentials were measured using the Malvern M3-PALS method and averaged from three measurements.

### 3.4. Luminescence Spectroscopy

Luminescence spectra were recorded on a Perkin Elmer LS 55 (Perkin Elmer, Waltham, MA, USA) at an excitation wavelength of 358 nm (excitation gap—3 nm) and emission scan range of 400‒600 nm (emission gap—7 nm), the gain of the photo detector was 650 V, the scanning speed was 250 nm per minute. Quartz cuvettes with an optical path length of 10 mm were used. The cuvette was located in the front face position. Deionized water was used for preparation of the solutions. Absorption spectra of mixtures were recorded after 1 h incubation at 20 °C. Excitation and emission slits were equal to 2 nm for DAPI in the presence of Pillar[5]arenes **1** and **2** and to 1 nm in the presence of DTAC and SDS. The fluorescence spectra were recorded with the 3 × 10^−5^ M concentration for DAPI. The concentration of the DAPI was determined individually by 3 wavelengths by the Beer–Lambert law method according to the linear section of the intensity variation, in the range of dye concentration from 3 × 10^−5^ to 5 × 10^−6^ M.

### 3.5. Nanoparticle Tracking Analysis (NTA)

The size, concentration and movement of nanoparticles were determined using a NanoSight LM10 instrument (Malvern Instruments Ltd., Worcestershire, UK) equipped with a C11440-50B CMOS camera with an FL-280 scientific image sensor (Hamamatsu Photonics, Hamamatsu, Japan) as a detector. Measurements were carried out in a special cell for organic solvents with a modified angle of entry of the laser beam into the solution, a 405 nm laser (version cd, S/N 2990491) and a Kalrez sealing ring. An HH804 contact thermometer (Omega Engineering, Inc., Stamford, CT, USA) was used to determine the temperature in the cell throughout the experiment. NanoSight NTA 2.3 software (build 0033) was used to process the results.

Video recording of the samples was carried out in a constant flow of liquid, for greater statistics. The dosing rate was 10 μL/min. From 3 to 5 video recordings were recorded, from 30 to 240 s long, depending on the number of particles in the solution. The first measurements to fix all visible particles in the solution were carried out without an extinguishing filter, followed by a cutoff filter at 430 nm and 500 nm. Thus, particles with emissivity were recorded at wavelengths of more than 430 and 500 nm, respectively.

### 3.6. Molecular Docking

DINC 2.0 web server [71,72] was used to study molecular interactions between Pillar[5]arene **2** and ligand (DAPI). DINC is a parallelized meta-docking method for the incremental docking of ligands. The strategy of DINC involves incrementally docking overlapping fragments with a growing number of atoms, while maintaining the number of flexible bonds constant during this incremental process [72]. The grid size was set to ~30 Å × 30 Å × 30 Å xyz points and grid center was designated at dimensions (x, y, and z): 0.8, 0.8, and −6.4.

## 4. Conclusions

Thus, the novel self-assembling systems based on oppositely charged Pillar[5]arenes and surfactants for encapsulation of dye DAPI were created. The aggregation behavior of macrocycles synthesized and surfactants in the presence of Pillar[5]arenes functionalized by carboxy and ammonium terminal groups were studied using the set physical methods. It was shown that by varying the molar ratio in Pillar[5]arene-surfactant systems, it is possible to obtain various types of nanosystems: (1) host-guest complexes at equimolar ratio of Pillar[5]arene-surfactant and (2) interpolyelectrolyte complexes (IPECs), are self-assembled materials formed in aqueous medium by two oppositely charged polyelectrolytes (macrocycle and surfactant micelles). It has been suggested that interaction of Pillar[5]arenes with surfactants is predominantly driven by cooperative electrostatic interactions. Synthesized stoichiometric and non-stoichiometric IPECs specifically interact with a third component, imaging system agent, 4′,6-diamidino-2-phenylindole (DAPI). UV-vis, luminescent spectroscopy and molecular docking data show the structural feature of dye-loaded IPEC and key role of the electrostatic, π–π-stacking, cation–π interactions in their formation. The nanoparticle trajectory analysis detected that the use of IPEC formed by surfactants and Pillar[5]arenes leads to an increase in the association constant, which is responsible for the complete binding of the dye. This work elucidated the mechanism of self-assembly between Pillar[5]arenes and surfactants, providing additional information on features of dye-loaded IPEC and the key role of the electrostatic, π–π-stacking, cation–π interactions in their formation. Design of dye-loaded supramolecular Pillar[5]arene-surfactant systems will certainly be in demand in pharmaceuticals and biomedical diagnostics.

## Figures and Tables

**Figure 1 ijms-22-06038-f001:**
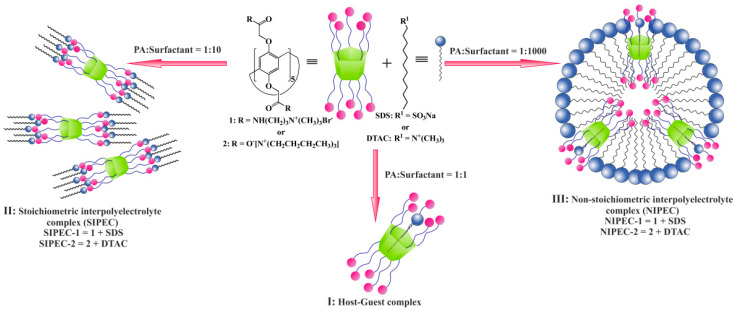
Schematic representation of the formation of different types of aggregates in different ratios of Pillar[5]arene-surfactant.

**Figure 2 ijms-22-06038-f002:**
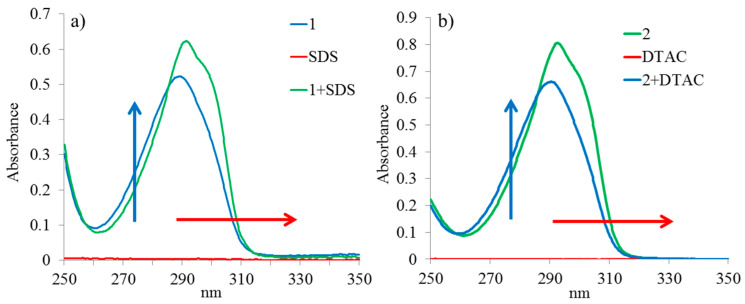
UV-vis absorption spectra: (**a**) Pillar[5]arene **1**/SDS in equimolar ratio in water (c_macrocycle_ = 3 × 10^−5^ M, c_guest_ = 3 × 10^−5^ M); (**b**) Pillar[5]arene **2**/DTAC at 1:1 ratio in water (c_macrocycle_ = 3 × 10^−5^ M, c_guest_ = 3 × 10^−5^ M).

**Figure 3 ijms-22-06038-f003:**
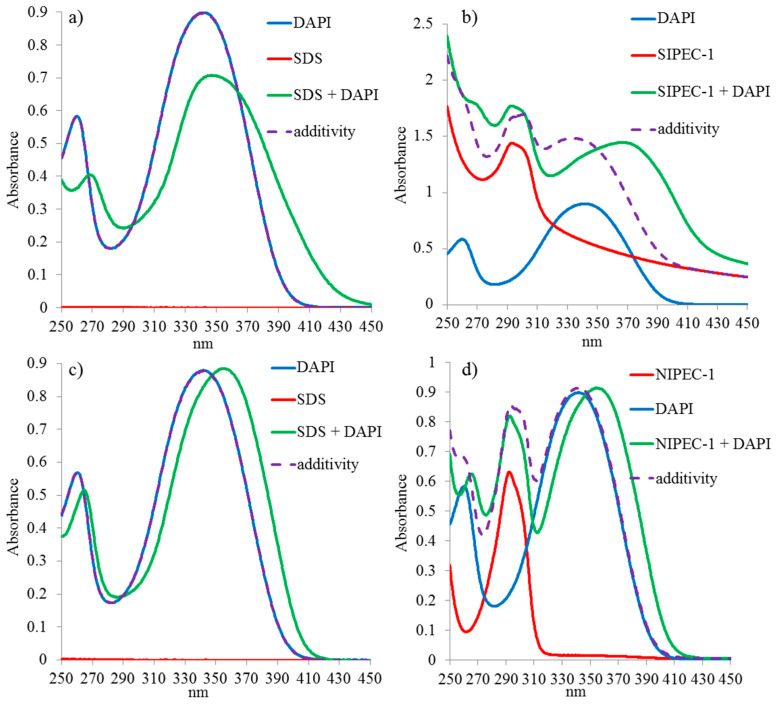
UV-vis absorption spectra: (**a**) SDS (c = 3 × 10^−3^ M) + DAPI (3 × 10^−5^ M); (**b**) **SIPEC-1** + DAPI (c = 3 × 10^−5^ M); (**c**) SDS (c = 3 × 10^−2^ M) + DAPI (c = 3 × 10^−5^ M); (**d**) **NIPEC-1** + DAPI (c = 3 × 10^−5^ M).

**Figure 4 ijms-22-06038-f004:**
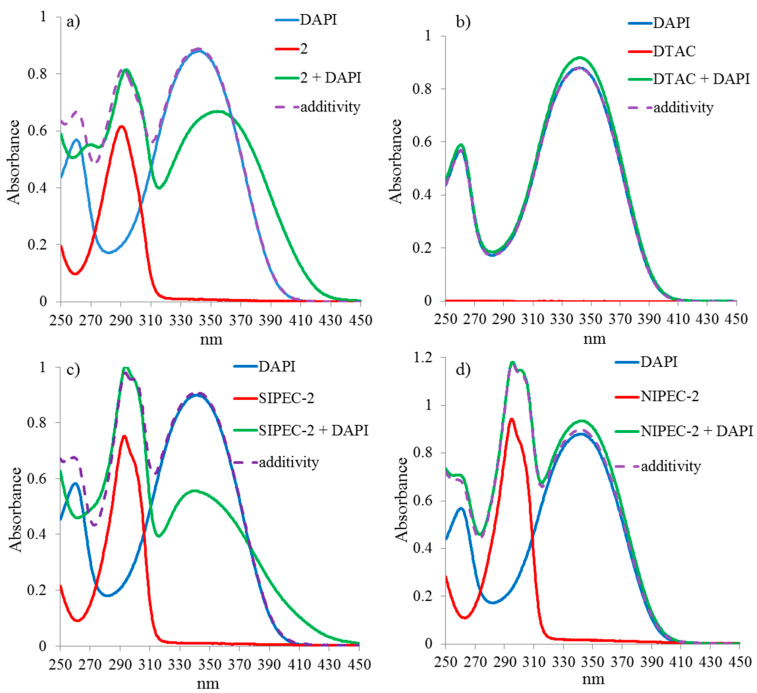
UV-vis absorption spectra: (**a**) Pillar[5]arene **2** (c = 3 × 10^−5^ M) + DAPI (c = 3 × 10^−5^ M); (**b**) DTAC (c = 3 × 10^−4^ M) + DAPI (c = 3 × 10^−5^ M); (**c**) **SIPEC-2** + DAPI (c = 3 × 10^−5^ M); (**d**) **NIPEC-2** + DAPI (c = 3 × 10^−5^ M).

**Figure 5 ijms-22-06038-f005:**
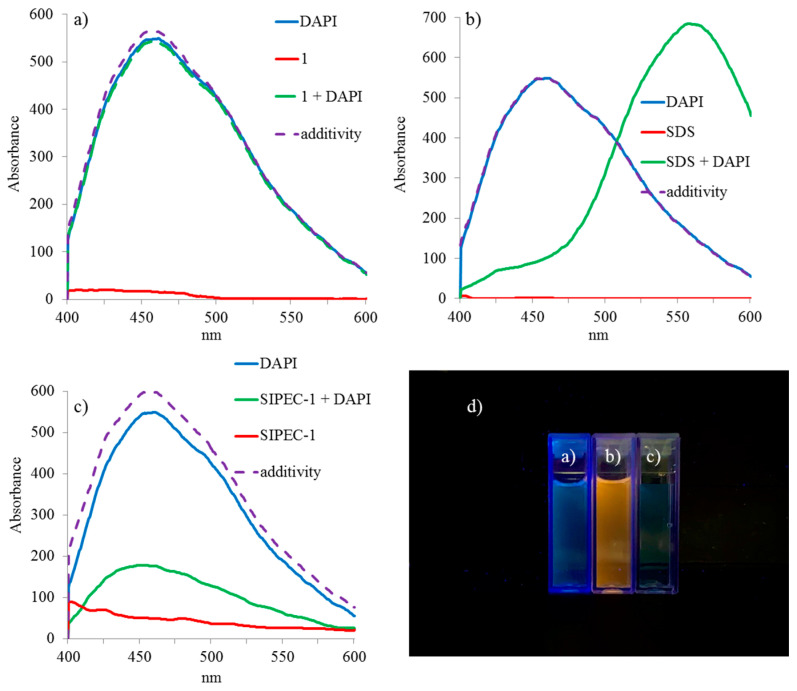
Luminescence spectra: (**a**) Pillar[5]arene **1** (c = 3 × 10^−5^ M) + DAPI (c = 3 × 10^−5^ M); (**b**) SDS (c = 3 × 10^−3^ M) + DAPI (c = 3 × 10^−5^ M); (**c**) **SIPEC-1** + DAPI (c = 3 × 10^−5^ M); (**d**) images of solutions a), (b), c).

**Figure 6 ijms-22-06038-f006:**
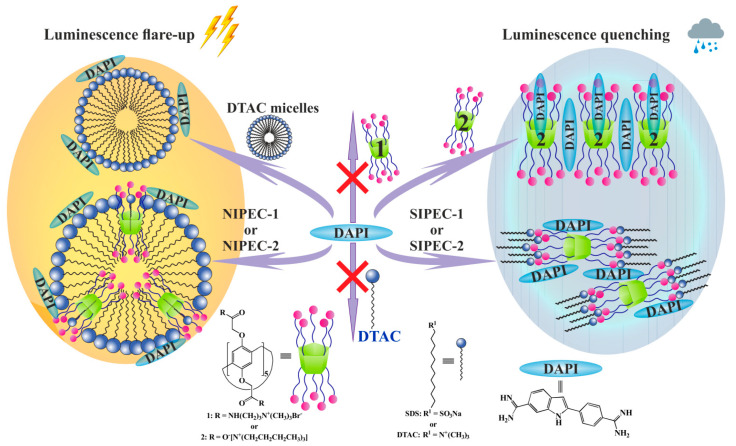
Schematic representation of DAPI interaction with macrocycles **1** and **2**, surfactants, IPEC based on macrocycles and surfactants.

**Figure 7 ijms-22-06038-f007:**
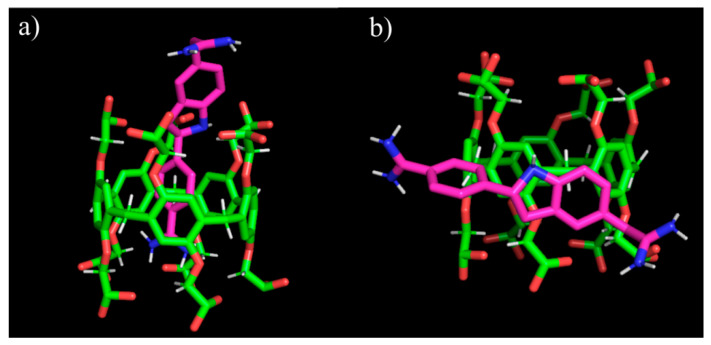
View of the geometry optimized complex formed by macrocycle **2** and 4′,6-diamidino-2-phenylindole (DAPI) calculated by DINC 2.0 web server: (**a**) the dye is inside the cavity; (**b**) the dye is outside the cavity.

**Figure 8 ijms-22-06038-f008:**
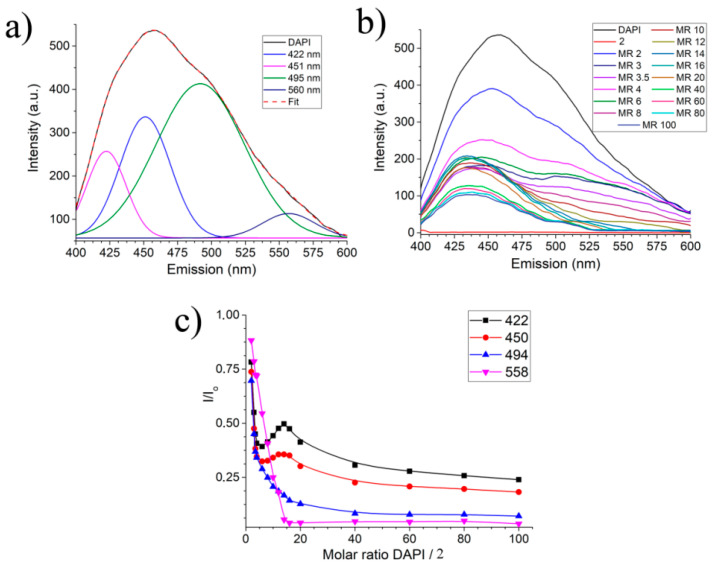
(**a**) DAPI luminescence spectrum with deconvolution of emission band (RMSE = 0.9996); (**b**) titration of DAPI by compound **2** indicating molar ratios c_m_(DAPI)/c_m_(**2**); (**c**) titration of DAPI by compound **2** in I/I_0_ format at different emission wavelengths.

**Figure 9 ijms-22-06038-f009:**
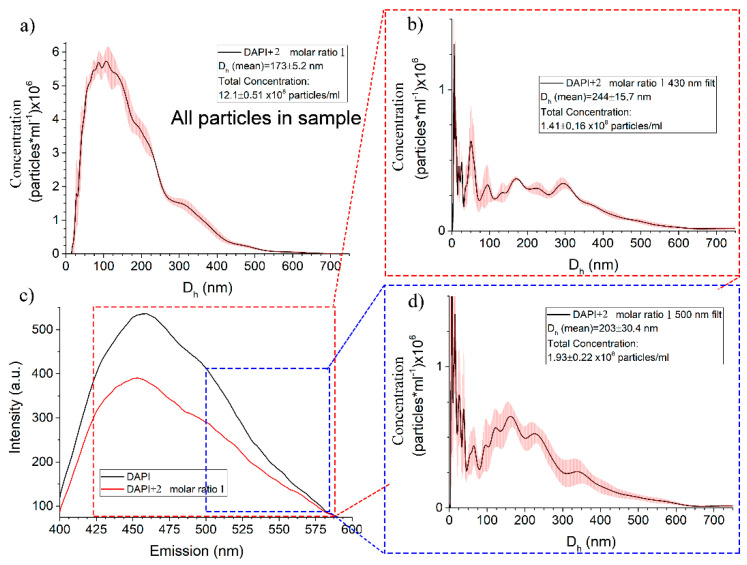
(**a**) Dependence of the concentration of particles on their size in a DAPI/**2** solution at a 1:1 molar ratio; (**b**) Dependence of the concentration of particles on their size in a solution of DAPI/**2** at a 1:1 molar ratio using a cutoff filter at 430 nm; (**c**) Luminescence spectrum of DAPI (c = 3 × 10^−5^ M) and DAPI/**2** at a 1:1 molar ratio; (**d**) Dependence of the concentration of particles on their size in a solution of DAPI/**2** at a 1:1 molar ratio using a cutoff filter at 500 nm.

**Figure 10 ijms-22-06038-f010:**
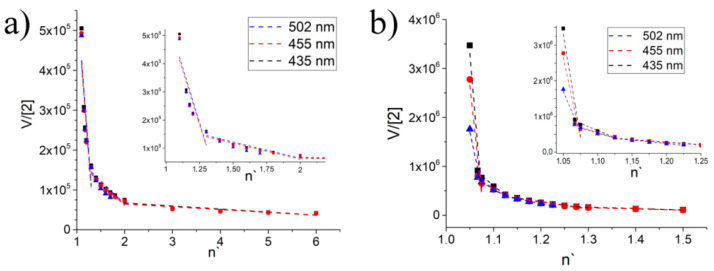
Determination of linearity sections using the Scatchard method: (**a**) relative to the equilibrium concentration of Pillar[5]arene **2**; (**b**) relative to the equilibrium concentration of macrocycle **2** in **SIPEC-2**/DAPI system.

**Table 1 ijms-22-06038-t001:** Calculated *K_a_* for the **2**/DAPI and **SIPEC-2**/DAPI systems.

K_a_	Emission 2/DAPI			Emission SIPEC-2/DAPI		
	435 nm	455 nm	500 nm	435 nm	455 nm	500 nm
1	1.56 × 10^6^	1.52 × 10^6^	1.50 × 10^6^	1.14 × 10^8^	8.98 × 10^8^	4.52 × 10^8^
2	1.15 × 10^5^	1.14 × 10^5^	1.15 × 10^5^	7.69 × 10^6^	6.11 × 10^6^	5.73 × 10^6^
3	7.57 × 10^3^	7.08 × 10^3^	N/A	1.43 × 10^6^	1.41 × 10^6^	1.94 × 10^6^
4	N/A	N/A	N/A	2.89 × 10^5^	2.42 × 10^5^	N/A

N/A: not applicable.

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
