# Peer review of "Self-Assembling Systems Based on Pillar[5]arenes and Surfactants for Encapsulation of Diagnostic Dye DAPI"

_ijms, 2021, doi:10.3390/ijms22116038_

Round 1
Reviewer 1 Report
Remarks about the manuscript:
The authors have presented "Novel self-assembling systems based on pillar[5]arenes and two surfactants for encapsulation of diagnostic dye DAPI". It is an exciting topic, self-assembly system of two surfactant molecule and their possible application in diagnostic role with DAPI. The data presented in this manuscript is excellent, with a proper explanation through various characterization techniques. Especially the characterization techniques and description about complexes such as UV-Vis, Photoluminescence, and DLS are great.
While the present manuscript contained most of the information required for the publication in the IJMS journal, one must think about the safety of this complex through proper cytotoxicity study before promises it for pharmaceutical and diagnostic applications.
In my view, this manuscript (ID: IJMS-1246856) can be accepted after considering the following comments:
- Authors have mentioned DAPI as an antiparasitic, antibiotic, antiviral, and antitumor agent [64,65], while both given references didn't perform such a study (page 2, line 72-74). The author should cross-check or add some relevant references about this claim.
- It would be better to study at least one pharmaceutical molecule to understand this complex application in biomedical science.
- Cytotoxic study of this self-assemble complex would be great as both the compound used for assembly are surfactant.
- Please Check the mentioned figure number (page 10, line 338)
- The authors have mentioned FTIR study in the material section but spectrum or description in the text (page 13, line 409).
- Does this complex still work if you want to encapsulate some charged particles for pharmaceutical or diagnostic purposes? Such as DNA or Peptide molecules?
Author Response
- Authors have mentioned DAPI as an antiparasitic, antibiotic, antiviral, and antitumor agent [64,65], while both given references didn't perform such a study (page 2, line 72-74). The author should cross-check or add some relevant references about this claim.
We cross checked some relevant references and corrected references.
- It would be better to study at least one pharmaceutical molecule to understand this complex application in biomedical science.
The proposal put forward by the Reviewer is extremely interesting and exciting. The study of the obtained IPECs with some drugs / pharmaceutical molecules is planned in the near future in order to develop the research topic. In this work, we used DAPI because its advantage is that the molecule is used both as a drug (antiparasitic, antibiotic, antiviral and antitumor agent) and as a fluorescent label for staining of DNA in fluorescence microscopy.
- Cytotoxic study of this self-assemble complex would be great as both the compound used for assembly are surfactant.
The biological activity of both complexes and components of this system will be investigated under the following works, in particular, in works on the inclusion of drugs and antioxidants. Here, for the first time, we showed approaches to obtaining such systems, because earlier in the literature IPEC based on pillar[5]arenes and surfactants did not exist.
- Please Check the mentioned figure number (page 10, line 338).
Line 338: “Figure 8” has been changed to “Figure 9”.
- The authors have mentioned FTIR study in the material section but spectrum or description in the text (page 13, line 409).
This is an unfortunate mistake.
Both studied compounds were synthesized according to literature methods. 1H NMR spectroscopy as well as measurement of the melting points were used to confirm their structure. In view of this, the first paragraph in 3.1. General was changed. The sentence “1H NMR and 13C NMR spectra were obtained... (13C{1H} 100 MHz and 1H 400 MHz)” was changed to “1H NMR spectrum was obtained…(1H 400 MHz)”. The sentences “The FTIR ATR spectra were recorded on the Spectrum 400 FT-IR spectrometer (Perkin Elmer Inc, Waltham, MA, USA) with a Diamond KRS-5 attenuated total internal reflectance attachment (resolution 0.5 cm−1, accumulation of 64 scans, recording time 16 s in the wavelength range 400–4000 cm−1). Electrospray ionization (ESI) mass spectra were measured on an AmazonX mass spectrometer (Bruker Daltonik GmbH, Germany) in positive ion mode in the m/z range of 100—2800. The capillary voltage was –4500 V; nitrogen as a nebulizer gas, 300 °C, the flow rate was 10 L min–1. The samples were dissolved in acetonitrile at a concentration of 10–6 g L–1. The data were processed with the DataAnalysis 4.0 software (Bruker Daltonik GmbH, Germany). Elemental analysis was performed on Perkin–Elmer 2400 Series II instruments (Perkin Elmer, Waltham, MA, USA)” were deleted.
- Does this complex still work if you want to encapsulate some charged particles for pharmaceutical or diagnostic purposes? Such as DNA or Peptide molecules?
IPECs are great for binding charged particles like DNA or peptide molecules. IPECs based on macrocyclic compounds (thiacalix[4]arene and pillar[5]arene) showed effective packaging of high-molecular DNA from calf thymus [DOI: 10.3390/nano10040777]. A. Zaichenko et al. obtained amphiphilic block-copolymer forming stable micelles and interpolyelectrolyte complexes with DNA for efficient gene delivery [DOI: 10.1080/00914037.2020.1740988]. This is an interesting and promising direction for the creation of therapeutic drugs without the drugs, for example, based on proteins. In our next works, we plan to develop in this direction.
Reviewer 2 Report
In this contribution by Nazarova and co-workers, the authors developed novel self-assembling systems based on oppositely charged pillar[5]arenes and surfactants for encapsulation of diagnostic dye DAPI. The results are interesting and potentially attractive to the readership of IJMS. I recommend it for publication after the following points are addressed.
- Line 32-33, one recent interesting review related to supramolecular drug delivery should be included to support this statement.
- Table 1, please use the typical three-lines table.
- Typical size distribution from DLS measurements for different self-assembling systems should be added.
- There is no microscopy result in this study to show the actual self-assembling structures.
Author Response
- Line 32-33, one recent interesting review related to supramolecular drug delivery should be included to support this statement.
The reference to a review on supramolecular drug delivery has been included.
- Table 1, please use the typical three-lines table.
We corrected the Table 1.
- Typical size distribution from DLS measurements for different self-assembling systems should be added.
Typical size distribution for different self-assembling systems has been added. Table S1 in supplementary information also has been changed.
- - There is no microscopy result in this study to show the actual self-assembling structures.
We plan to use the systems as therapeutic agents and fluorescent labels. This implies their use as solutions and powders that will be dissolved in physiological fluids. Therefore, the primary task was to study their formation and behavior in aqua medium. A set of methods (DLS, UV-vis, luminescent spectroscopy, NTA) was applied for this purpose. Moreover, IPEC aqueous solutions have been comprehensively characterized. To use IPEC as coatings for biochemical sensors, we would definitely use the microscopy method to control the quality of the coating.